# Cross-reactive Dengue virus-specific CD8+ T cells protect against Zika virus during pregnancy

Jose Angel Regla-Nava[1], Annie Elong Ngono[1], Karla M. Viramontes[1], Anh-Thy Huynh[1], Ying-Ting Wang[1], Anh-Viet T. Nguyen[1], Rebecca Salgado[1], Anila Mamidi[1], Kenneth Kim[1], Michael S. Diamond[2] & Sujan Shresta[1,3]

As Zika virus (ZIKV) emerges into Dengue virus (DENV)-endemic areas, cases of ZIKV infection in DENV-immune pregnant women may rise. Here we show that prior DENV immunity affects maternal and fetal ZIKV infection in pregnancy using sequential DENV and ZIKV infection models. Fetuses in ZIKV-infected DENV-immune dams were normal sized, whereas fetal demise occurred in non-immune dams. Moreover, reduced ZIKV RNA is present in the placenta and fetuses of ZIKV-infected DENV-immune dams. DENV cross-reactive CD8+ T cells expand in the maternal spleen and decidua of ZIKV-infected dams, their depletion increases ZIKV infection in the placenta and fetus, and results in fetal demise. The inducement of cross-reactive CD8+ T cells via peptide immunization or adoptive transfer results in decreased ZIKV infection in the placenta. Prior DENV immunity can protect against ZIKV infection during pregnancy in mice, and CD8+ T cells are sufficient for this cross-protection. This has implications for understanding the natural history of ZIKV in DENV-endemic areas and the development of optimal ZIKV vaccines.

[1] Division of Inflammation Biology, La Jolla Institute for Allergy & Immunology, 92037 La Jolla, CA, USA. [2] Department of Medicine, Molecular Microbiology, Pathology, and Immunology, Center for Human Immunology and Immunotherapy Programs, Washington University School of Medicine, 63110 St. Louis, MO, USA. [3] Department of Medicine, School of Medicine, University of California, San Diego, 92093 La Jolla, CA, USA. These authors contributed equally: Jose Angel Regla-Nava, Annie Elong Ngono.  Correspondence and requests for materials should be addressed to S.S. (email: sujan@lji.org)

Zika virus (ZIKV) is a positive-stranded, enveloped, RNA flavivirus in the *Flaviviridae* family that is transmitted by *Aedes* species mosquitoes and sexual contact. ZIKV was first isolated in 1947 from a sentinel rhesus macaque in Uganda, and for decades, sporadic human case reports in Africa and Asia were associated with a self-limiting febrile illness. Outbreaks of ZIKV infection beyond its original range were reported in 2007 in Micronesia and from 2013 to 2014 in French Polynesia, where infection was associated with development of Guillain–Barré syndrome (GBS)[1]. Recently, there was a major epidemic of ZIKV in the Western Hemisphere, which also was associated with GBS. Additionally, infection of pregnant women was confirmed to cause congenital ZIKV syndrome, which includes microcephaly and other birth defects[2,3].

A successful pregnancy requires the maternal immune system to recognize and tolerate fetal tissues. Nonetheless, pregnant mammals must still mount robust immune response to pathogens[4–6]. Some pathogens including ZIKV ostensibly evade the immune system and breach the maternal–fetal interface. The primary barrier between the maternal and fetal compartments during pregnancy is the fetally derived placenta that is adjacent to and intercalated with the maternal decidua. Fetal macrophages (Hofbauer cells), placental fibroblasts, fetal endothelial cells and syncytiotrophoblasts, together with decidual stromal cells, macrophages, and lymphocytes of maternal origin, protect the fetus from pathogens present in maternal blood[7–9]. Several studies in animal models have demonstrated vertical transmission of ZIKV and its tropism for placental cells, including trophoblasts, endothelial cells, and macrophages[10–15]. Once ZIKV crosses the placental barrier, it can infect neuronal progenitor cells in the fetal brain[10,12,16–18].

ZIKV and the closely related flavivirus DENV co-circulate in the same geographic ranges and are transmitted by the same mosquitoes. ZIKV and the four serotypes of dengue virus (DENV1–4) share 55.1–56.3% amino acid sequence identity. The adaptive immune response to DENV and its roles in protection versus pathogenesis is complex and remains incompletely understood[19]. Epidemiological data indicate that following primary infection by one DENV serotype, a second infection with a different DENV serotype may lead to a more severe form of dengue disease, revealing potential roles for antibodies (Abs) and T cells in DENV pathogenesis. Two hypotheses have been proposed to explain this phenomenon: Ab-dependent enhancement (ADE) and T cell original antigenic sin (TOAS). Many studies support the ADE model[20–24] while the role for T cells remains less clear. Indeed, recent data indicate protective roles for serotype-specific and cross-reactive T cells against DENV infection in humans[25–30] and mice[31–37]. The role of T cells in ZIKV immunity also has been explored in animal models. In non-human primates, the peak of the CD8[+] T cell activation correlates with ZIKV RNA reduction, suggesting a protective role for CD8[+] T cells in controlling ZIKV replication[38]. In mice, CD8[+] T cells expand, exhibit high cytolytic activity, and mediate viral clearance[39]. Based on amino acid sequence and structural similarities between DENV and ZIKV, many groups have shown cross-reactivity between DENV and ZIKV in both humoral[40–45] and cellular responses[46–49]. One study in non-human primates showed that prior DENV exposure resulted in a reduction in the duration of ZIKV viremia in DENV-immune animals, suggesting cross-protection[50], although another group reported more neutral effects of DENV immunity on ZIKV infection and disease pathogenesis[51]. Studies in mice have shown that DENV/ZIKV cross-reactive Abs can enhance ZIKV pathogenesis[41,42], whereas DENV/ZIKV cross-reactive CD8[+] T cells protect against ZIKV infection[46,49,52]. However, no study has evaluated (i) how prior DENV exposure influences maternal and fetal outcome of ZIKV infection in pregnancy and (ii) the contribution of CD8[+] T cells to protect against or pathogenesis of ZIKV infection during pregnancy.

Accordingly, we investigated the outcomes of ZIKV infection during pregnancy in DENV-immune and non-immune mice using short-term sequential infection models. Prior exposure to DENV conferred protection against maternal and fetal ZIKV infection, and DENV-immune CD8[+] T cells were required for this cross-protection. The cross-reactive antigen-specific T cells in the maternal spleen and maternal–fetal interface exhibited a polyfunctional phenotype and expressed multiple cytokine and cytotoxicity markers. Moreover, in the absence of prior DENV immunity, induction by peptide immunization or adoptive transfer of cross-reactive CD8[+] T cells was sufficient to protect against ZIKV infection in the setting of pregnancy. These results demonstrate that DENV/ZIKV cross-reactive CD8[+] T cells can limit trans-placental transmission of ZIKV, and suggest that vaccines and therapeutics that optimize this response may minimize vertical transmission and fetal injury.

## Results

**DENV2-CD8[+] T cells protect against fetal ZIKV infection.** We recently demonstrated that DENV-elicited CD8[+] T cells mediated cross-protection against subsequent ZIKV infection in adult male and female *Ifnar1*[−/−] mice[49]. Previously, fetal growth restriction and demise have been observed in *Ifnar1*[−/−] pregnant mice following ZIKV infection[10,11]. We first assessed whether the ZIKV-induced fetal demise phenotype was affected by the route of inoculation: we compared retro-orbital (RO) and subcutaneous (via footpad) (SC) routes of ZIKV infection. *Ifnar1*[−/−] dams were crossed with WT males and then challenged with $10^4$ focus forming units (FFU) of ZIKV strain FSS13025 (2010 Cambodian isolate) on embryonic day 7.5 (E7.5) and killed 7 days later (E14.5). After challenge with this ZIKV strain, fetuses exhibited more consistent and severe resorption phenotypes upon RO than SC route of infection (Supplementary Fig. 1). We therefore used the RO route of ZIKV inoculation for the present study to more clearly assess protection. To begin to evaluate the influence of prior DENV immunity on subsequent ZIKV infection during pregnancy, we utilized our published model of sequential DENV and ZIKV infection in which mice were primed with DENV2 strain S221 for 30 days prior to ZIKV challenge[49]. Non-immune and DENV-immune *Ifnar1*[−/−] pregnant mice were inoculated with $10^4$ FFU of ZIKV FSS13025 on E7.5 and killed 7 days later (E14.5). In the non-immune group, fetal resorption was observed after ZIKV infection in all mice regardless of treatment with an isotype control or anti-CD8 Ab (Fig. 1a, b). Decreased fetal weight (Fig. 1a) and size (Fig. 1b) at E14.5 were consistently observed in the ZIKV-infected non-immune group. Remarkably, DENV-immune dams treated with isotype control Ab had normal fetuses that were similar in size to uninfected, naive control dams. Fetal resorption was observed in DENV2-immune mice only in the anti-CD8 Ab-treated group (Fig. 1a, b, e). These results indicate that prior DENV immunity prevents fetal resorption induced by ZIKV infection during pregnancy, and CD8[+] T cells contribute to DENV-immune-mediated protection against ZIKV in *Ifnar1*[−/−] pregnant mice.

As CD4[+] T cell help may be required for development of an optimal CD8[+] T cell response, we examined their role in DENV-immune-mediated protection against ZIKV during pregnancy. Non-immune and DENV2-immune dams were depleted of CD4[+] T cells or both CD4[+] and CD8[+] T cells via treatment with cell-depleting anti-CD4 or anti-CD4 plus anti-CD8 Abs. Fetuses undergoing resorption were seen in ZIKV-challenged, non-immune groups treated with anti-CD4 Ab alone or both anti-

CD4 and anti-CD8 Abs (Fig. 1c, d). However, with prior DENV2 immunity, an intermediate phenotype with 47% of viable fetuses was found in mice treated with anti-CD4 Ab alone as compared with nearly 100% resorption in the group treated with both anti-CD4 and anti-CD8 Abs (Fig. 1c, d, f). These results suggest that

cross-reactive CD4$^+$ and CD8$^+$ T cells have subordinate and dominant roles, respectively, in mediating DENV-immune protection against ZIKV-induced fetal damage.

We next determined the impact of prior DENV immunity on ZIKV burden in maternal tissues 7 days after inoculation at

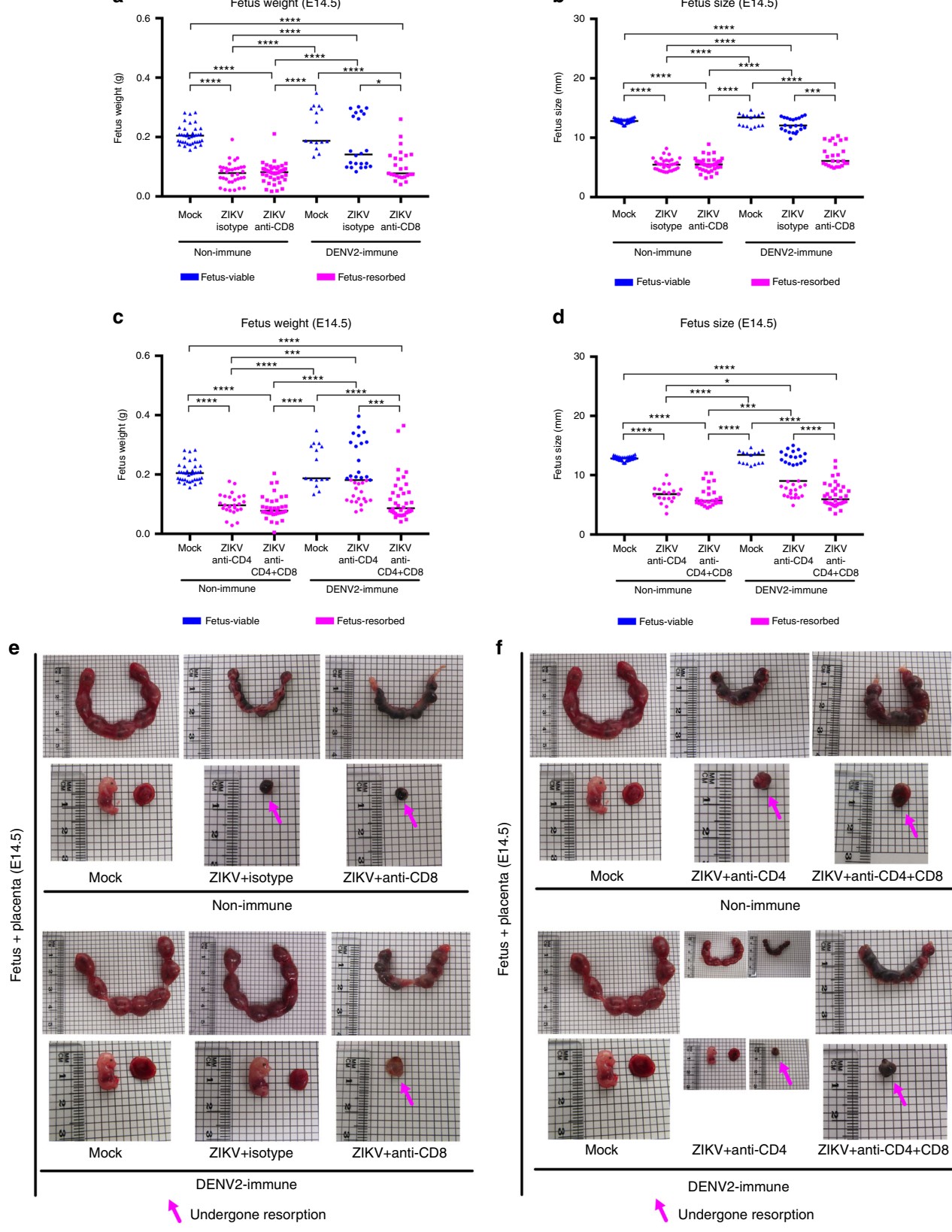

E14.5. In all cases, dams treated with anti-CD8 Ab had increased ZIKV RNA levels in the serum, brain, and spleen compared with those treated with isotype control Ab, and DENV immunity resulted in decreased viral RNA levels compared with the non-immune group (Supplementary Fig. 2a–c). In contrast, dams treated with anti-CD4 Ab had similar maternal tissue viral burdens as isotype control Ab-treated animals (Supplementary Fig. 2d–f). Comparison of anti-CD4 versus anti-CD8 Ab-treated groups also revealed that anti-CD8 but not anti-CD4 Ab treatment affected ZIKV RNA levels in maternal tissues (Supplementary Fig. 2g–i). Finally, no difference was observed between the group treated with anti-CD8 Ab alone versus that treated with both anti-CD4 and anti-CD8 Abs in the brain and spleen of both non-immune and DENV-immune mice and serum of non-immune mice, whereas a higher ZIKV RNA burden was observed in DENV-immune mice administered both anti-CD4 and anti-CD8 Abs than in those treated with anti-CD8 Ab alone (Supplementary Fig. 2j–l). Collectively, these results show a key role for CD8[+] T cells, with a more limited requirement for CD4[+] T cells, in DENV-immune-mediated cross-protection of ZIKV infection in maternal tissues from $Ifnar1^{-/-}$ mice.

**DENV2-CD8[+] T cells limit fetal ZIKV infection in WT mice.** To confirm and extend these findings, we utilized a published model of ZIKV vertical transmission in WT mice with transient Ifnar1 blockade[10]. Pretreatment with the Ifnar1-blocking Ab MAR1-5A3[53] allows flaviviruses to replicate in WT mice without substantively impacting CD8[+] T cell differentiation into effector and memory cells[54]. WT C57BL/6 female mice were administered anti-Ifnar1 Ab 1 day before infection with DENV2, as DENV cannot inhibit type I interferon production and signaling in mouse cells, unlike in human cells[19]. Thirty days after DENV2 priming, mice were mated with male sires, followed by treatment of DENV2-immune and non-immune WT dams with anti-Ifnar1 Ab 1 day prior to ZIKV challenge on E7.5. Seven days later, at day E14.5, fetal weight, size, and characteristics were recorded, and maternal and fetal tissues were harvested. As seen in the $Ifnar1^{-/-}$ mouse model, decreased fetal weight and size were observed in ZIKV-infected non-immune mice, whereas prior DENV immunity prevented fetal growth restriction with the fetal size and weight comparable to the mock-infected control group (Fig. 2a, b). Again, fetal growth restriction and resorption were observed in DENV-immune mice treated with anti-CD8 Ab (Fig. 2a, b and Supplementary Fig. 3). These results demonstrate that, in WT mice with transient Ifnar1 blockade, prior DENV immunity affords protection against ZIKV-induced fetal growth restriction in a CD8[+] T cell-dependent manner.

Analysis of viral burden revealed that ZIKV RNA was consistently present in maternal spleen, placenta with decidua, and fetal body, with reduced levels in DENV-immune mice relative to non-immune dams. Administration of anti-CD8 Ab abrogated the protective effect of DENV immunity, with significantly higher viral RNA levels detected in both maternal and fetal tissues compared to isotype control Ab-treated DENV-immune dams (Fig. 2c–h). The efficiency of anti-CD8 Ab-mediated CD8[+] T cell depletion in the spleen and decidua/placenta were >95% (Supplementary Fig. 4a, b). In DENV-immune dams, anti-CD8 Ab treatment resulted in a greater percentage of ZIKV infection in placentas with decidua and fetal heads and bodies compared with isotype control Ab treatment, with 100% versus 74% in placentas with decidua, 69% versus 37% in fetal heads, and 82% versus 44% in fetal bodies. The differences between the isotype and anti-CD8 Ab-treated DENV-immune mice were significant for all three tissues (Fig. 2i–k) (Two-sided Fisher's exact test: $p < 0.001$ for placenta with decidua, $p < 0.01$ for fetal head, and $p < 0.001$ for fetal body). As decreased maternal ZIKV viremia may lead to lower ZIKV levels in the maternal–fetal interface, we compared viral burden in maternal and fetal tissues of DENV-immune mice treated with isotype control versus anti-CD8 Ab at early time points after ZIKV challenge. At E9.5 and E10.5 (2 and 3 days after ZIKV challenge of E7.5 mothers), similar levels of ZIKV RNA were present in the maternal serum of DENV-immune mice treated with isotype control Ab or anti-CD8 (Fig. 3a). In contrast, higher ZIKV RNA levels were detected in the maternal brain and spleen only at E10.5 and in the placenta with decidua at both E9.5 and E10.5 in anti-CD8 Ab-treated than isotype control Ab-treated mice (Fig. 3b–d). Taken together, these data suggest that in WT mice with transient Ifnar1 blockade, prior DENV immunity controls ZIKV infection in both maternal and fetal tissues via DENV-exposed memory CD8[+] T cells. However, early during infection, ZIKV levels in circulation are not impacted by these T cells; once blood-borne ZIKV spreads and replicates in distal sites, including the maternal–fetal interface, DENV-elicited memory CD8[+] T cells assume an important role in limiting ZIKV infection of tissues.

Next, to confirm the role of CD8[+] T cells in protecting against ZIKV infection during pregnancy in DENV2-immune mice, gene-deficient mice ($Cd8^{-/-}$) lacking CD8[+] T cells were used. DENV-immune $Cd8^{-/-}$ mice were treated with Ifnar1-blocking Ab, followed by challenge with ZIKV at E7.5 and harvest at E14.5. The fetuses exhibited both growth restriction and resorption phenotypes, similar to that of WT mice treated with depleting anti-CD8 Ab (Fig. 4a). Decreased fetal weight (Fig. 4b) and size (Fig. 4c) at E14.5 were observed in DENV2-immune $Cd8^{-/-}$

**Fig. 1** Phenotype of fetuses, fetal weight, and size during maternal ZIKV infection in $Ifnar1^{-/-}$ mice with or without depletion of CD8[+], CD4[+], or both CD4[+] and CD8[+] T cells. Non-immune and DENV2-immune $Ifnar1^{-/-}$ dams that were administered isotype control Ab (Isotype), anti-CD4 Ab (Anti-CD4), anti-CD8 Ab (Anti-CD8), or both Abs (Anti-CD4 + CD8) were inoculated via a retro-orbital route (RO) at embryonic day 7.5 (E7.5) with $10^4$ FFU of ZIKV FSS13025 or 10% FBS-PBS as Mock. To generate DENV2-immune mice, mice were inoculated via an intraperitoneal route (IP) with $10^3$ FFU of DENV2 strain S221 for 30 days prior to mating. **a–d** Fetal body weight (**a, c**) and size (**b, d**) were measured at E14.5. **e, f** Representative pictures from uterus, fetus, and placenta at E14.5 from ZIKV-infected non-immune or DENV2-immune dams with (right) or without (center) anti-CD8 Ab administration (**e**) or dams treated with anti-CD4 (center) or anti-CD4 + CD8 Ab (right) administration (**f**). Magenta arrows indicate the presence of fetuses having undergone resorption. $n = 38$ fetuses from five separate mothers (non-immune-Mock), $n = 34$ fetuses from four separate mothers (non-immune-ZIKV + isotype), $n = 43$ fetuses from five separate mothers (non-immune-ZIKV + Anti-CD8), $n = 15$ fetuses from three separate mothers (DENV2-immune-Mock), $n = 22$ fetuses from three separate mothers (DENV2-immune-ZIKV + isotype), $n = 25$ fetuses from three separate mothers (DENV2-immune-ZIKV + Anti-CD8), $n = 26$ fetuses from four separate mothers (non-immune-ZIKV + Anti-CD4), $n = 36$ fetuses from four separate mothers (non-immune-ZIKV + Anti-CD4 + CD8), $n = 35$ fetuses from four separate mothers (DENV2-immune-ZIKV + Anti-CD4), and $n = 38$ fetuses from five separate mothers (DENV2-immune-ZIKV + Anti-CD4 + CD8). Weight and size were determined individually on the residual placenta if fetal resorption was observed. Data were pooled from two independent experiments. Kruskal–Wallis test was used, and data are expressed as median. *$p < 0.05$, ***$p < 0.001$, ****$p < 0.0001$

mice, similar to DENV2-immune WT mice treated with depleting anti-CD8 Ab (Fig. 4b, c). Thus, both loss-of-function models for murine CD8+ T cells demonstrated that DENV2-primed CD8+ T cells are responsible for mediating cross-protection against ZIKV infection during pregnancy.

To begin to define the duration of cross-protection mediated by DENV2-primed CD8+ T cells against ZIKV infection during pregnancy, we evaluated the viral phenotype in dams that were primed with DENV2 for 80 days prior to ZIKV challenge. No significant differences in fetal weight and size were observed between non-immune and DENV2-immune dams with ZIKV infection (Supplementary Fig. 5a–c). Although reduced levels of ZIKV RNA were present in maternal spleens of DENV-immune relative to non-immune dams (Supplementary Fig. 5d), no difference in viral burden was observed in the placenta/decidua of DENV-immune and non-immune dams (Supplementary Fig. 5e).

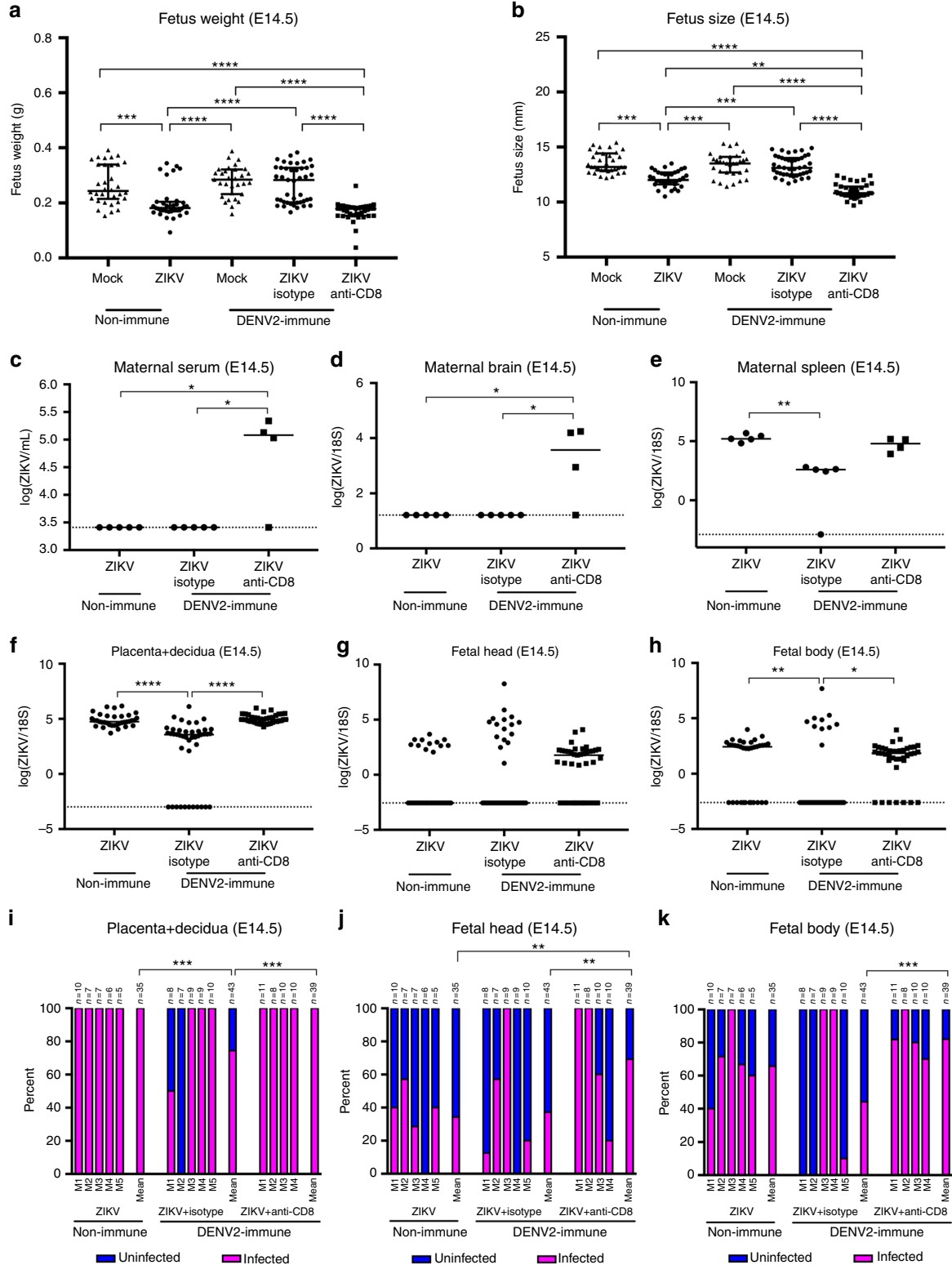

Thus, and analogous to that seen in natural infections with flaviviruses in humans[55–59], the protection mediated by cross-reactive CD8+ T cells against ZIKV infection is short term in nature.

**CD8+ T cells in the maternal spleen of DENV-immune WT mice.** We recently identified five H-2$^b$-restricted ZIKV-derived CD8+ T cell epitopes (prM$_{169–177}$, E$_{294–302}$, E$_{297–305}$, NS3$_{1866–1874}$, and NS5$_{2783–2792}$) that were cross-reactive with those induced in DENV2-infected mice[49]. To understand the contribution of DENV2-elicited CD8+ T cells to protect against ZIKV infection during pregnancy in DENV-immune WT mice with transient Ifnar1 blockade, we assessed the quantity and phenotype of cross-reactive CD8+ T cells in the maternal spleen by performing intracellular cytokine staining (ICS) analysis. We first confirmed the presence of cross-reactive antigen-specific CD8+ T cells in DENV-immune dams 3 days after ZIKV challenge (E10.5); we chose this time point because day 3 post infection is too early for detection of the primary but not memory anti-ZIKV T cell response in adult male and virgin female mice[39,46]. The gating strategy used to identify cells of interest is illustrated in Fig. 5a. Both frequencies and numbers of cross-reactive epitope-specific CD8+ T cells that were CD44$^{hi}$ CD62L$^{low}$ (i.e., effector memory in DENV-immune and primary effectors in non-immune mice) and expressed IFNγ alone (Fig. 5b–d), both IFNγ and TNF (Fig. 5e–g), or granzyme B (Fig. 5h–j) were higher in DENV-immune than non-immune dams. These results indicated that prior DENV exposure elicited cross-reactive effector memory CD8+ T cell responses in the maternal spleen during subsequent ZIKV infection of WT dams with transient Ifnar1 blockade.

We next compared the five cross-reactive epitope-specific CD8+ T cell responses in the maternal spleen from non-immune versus DENV-immune dams on day 7 after ZIKV challenge (E14.5), when the primary CD8+ T cell response to ZIKV infection in non-immune animals should peak[39]. The frequencies but not numbers of three of the five epitope-specific CD44$^{high}$CD62L$^{low}$ effector memory and effector CD8+ T cells producing IFNγ were higher in DENV-immune than non-immune mice (Fig. 6a–c). In comparison, the frequencies of all five epitope-specific CD8+ T cells and the number of some epitope-specific CD8+ T cells producing both IFNγ and TNF were higher in DENV-immune than non-immune dams (Fig. 6d–f), and the percentages of all five epitope-specific and the number of four epitope-specific CD8+ T cells expressing granzyme B were greater in DENV2-immune than non-immune animals (Fig. 6g–i). These results revealed an increased polyfunctional nature of the cross-reactive CD8+ T cell response in DENV-immune relative to non-immune dams.

**CD8+ T cells in the decidua of DENV-immune WT mice.** CD8+ T cells are one of the key cell types that are present in the decidua, which are located on the maternal side of the placenta[60]. Therefore, we next addressed whether cross-reactive CD8+ T cells were located at the maternal–fetal interface on day 7 after ZIKV challenge of non-immune and DENV-immune WT mice. Although decidual T cells were rare compared to splenic T cells, perhaps due to epigenetic silencing of key chemokine genes that prevent influx of T cells to the decidua[61], we detected polyfunctional effector memory and effector CD8+ T cells in the decidua/placenta after restimulation with a mixture of the five cross-reactive epitopes (Fig. 7a). Although CD8+ T cells producing IFNγ alone or both IFNγ and TNF were present in both non-immune and DENV-immune dams, significantly greater numbers were evident in the decidua of DENV-immune than non-immune mice (Fig. 7b, c). Similarly, higher numbers of CD8+ T cells expressing granzyme B were present in DENV-immune compared to non-immune dams after ZIKV infection (Fig. 7d, e). Mating of females expressing a CD45.2 allele with congenic males expressing a CD45.1 allele revealed that >95% of antigen-specific CD8+ T cells in the placenta/decidua of ZIKV-infected dams were CD45.2+, indicating a maternal origin of cross-reactive CD8+ T cells at the maternal–fetal interface (Supplementary Fig. 6). Thus, maternal cross-reactive antigen-specific CD8+ T cells with polyfunctional phenotype were present in the decidua/placenta of ZIKV-challenged DENV-immune mice, whereas few antigen-specific CD8+ T cells were observed in the decidua/placenta of ZIKV-infected non-immune dams.

We next assessed the protective function of the cross-reactive CD8+ T cell response against ZIKV infection in the decidua/placenta by performing peptide immunization and adoptive transfer experiments. For peptide immunization experiments, mice were vaccinated with a pool of the five cross-reactive peptides or DMSO alone as the mock group. On day 7 after ZIKV challenge, we measured viral RNA levels by qRT-PCR. The quantity of ZIKV RNA was significantly decreased in the placenta with decidua of dams previously immunized with the cross-reactive peptides as compared to mock-immunized mice (Fig. 7f), indicating a protective function of cross-reactive viral antigen-specific CD8+ T cells against ZIKV infection of the placenta/decidua. Additionally, we performed adoptive transfer experiments in which DENV2-immune or non-immune CD8+ T cells were transferred into non-immune recipient dams prior to infection with ZIKV. The level of ZIKV RNA was significantly reduced in the placenta/decidua of dams that received DENV2-primed CD8+ T cells compared to non-immune CD8+ T cells (Fig. 7g). Thus, both peptide immunization and adoptive transfer studies demonstrated that cross-reactive CD8+ T cells are sufficient to control ZIKV infection in the placenta with decidua (Fig. 7f, g).

**Discussion**

As the number of ZIKV infections in pregnant women increases, more cases of congenital ZIKV syndrome likely will occur. As

**Fig. 2** Fetal weight and size and viral burden in Ifnar1-blocking Ab-treated WT dams with or without CD8+ T cell depletion. Non-immune and DENV2-immune WT dams that were administered isotype control Ab (Isotype) or anti-CD8 Ab (Anti-CD8) were inoculated RO at embryonic day 7.5 (E7.5) with 10$^4$ FFU of ZIKV FSS13025 or 10% FBS–PBS as Mock. All mice were injected IP with Ifnar1-blocking Ab on E6.5, 1 day prior to ZIKV challenge. DENV2-immune mice were generated after RO infection with 10$^4$ FFU of DENV2 strain S221 for 30 days prior to mating. **a** Fetus weight and **b** size at E14.5 were recorded. **c**–**h** ZIKV RNA levels were measured by qRT-PCR of tissues collected from dams (serum, brain, and spleen), placentas with decidua, and fetuses (head and body) at E14.5. **i**–**k** Percentages of ZIKV infection in placentas with decidua, fetal heads, and fetal bodies at E14.5 were calculated. $n = 34$ fetuses from six separate mothers (non-immune-Mock), $n = 35$ fetuses from five separate mothers (non-immune-ZIKV), $n = 31$ fetuses from four separate mothers (DENV2-immune-Mock), $n = 43$ fetuses from five separate mothers (DENV2-immune-ZIKV + isotype), and $n = 39$ fetuses from four separate mothers (DENV2-immune-ZIKV + Anti-CD8). Total numbers of the fetal and placental units obtained from each dam in each group are indicated above each bar. Data were pooled from two independent experiments. Data are expressed as median. *$p < 0.05$, **$p < 0.01$, ***$p < 0.001$, ****$p < 0.0001$. Kruskal–Wallis test was used for **a**–**h**, while two-sided Fisher's exact test was used for **i**–**k**

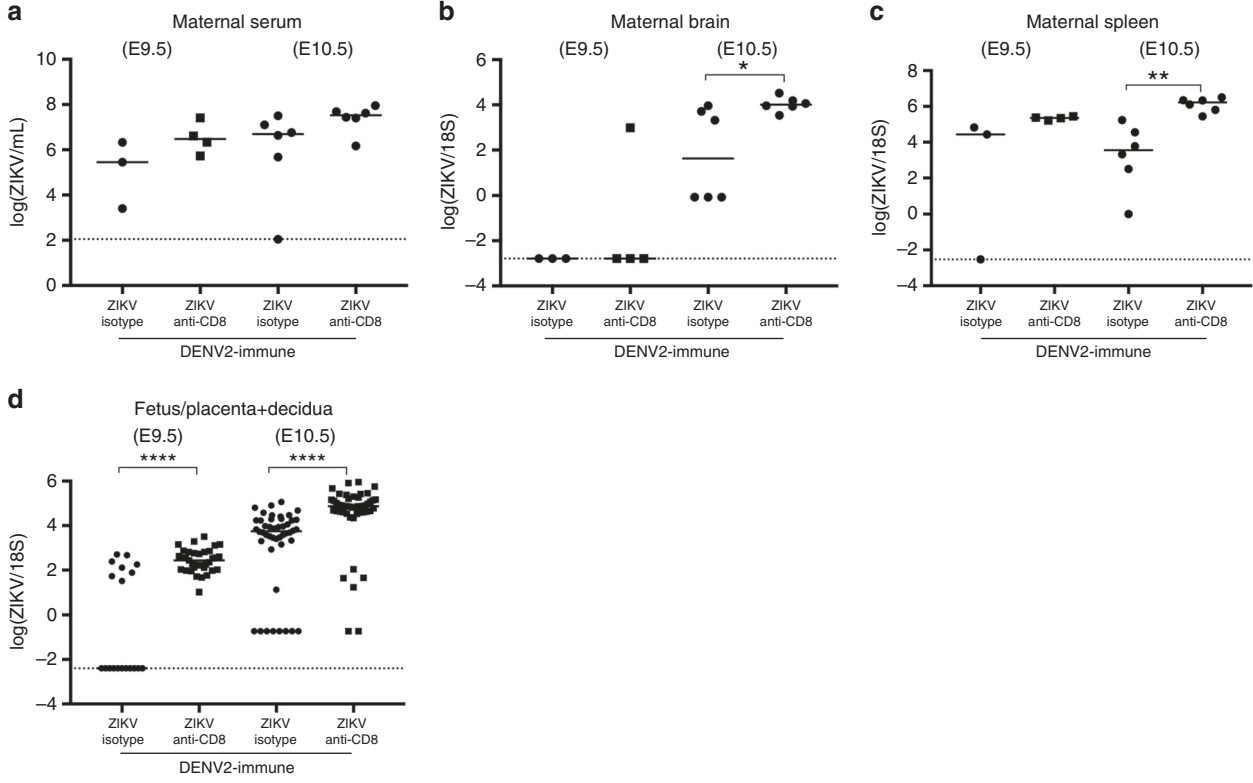

**Fig. 3** ZIKV burden in Ifnar1-blocking Ab-treated WT dams and their fetuses on days 2 and 3 after ZIKV infection. DENV2-immune WT dams were treated with anti-Ifnar1 mAb and challenged with ZIKV at E7.5 as described in Fig. 2. Mice were administered isotype control or anti-CD8 Ab, also as described in Fig. 2. On days 2 and 3 after ZIKV challenge (E9.5 and E10.5), ZIKV RNA levels in maternal **a** serum, **b** brain, and **c** spleen and **d** fetus + placenta + decidua were measured by qRT-PCR. $n = 19$ fetuses from three separate mothers (ZIKV + isotype) and $n = 35$ fetuses from four separate mothers (ZIKV + Anti-CD8) at E9.5. $n = 46$ fetuses from six separate mothers (ZIKV + isotype) and $n = 48$ fetuses from six separate mothers (ZIKV + Anti-CD8) at E10.5. Data were pooled from two independent experiments. Data are expressed as median. *$p < 0.05$, **$p < 0.01$, ****$p < 0.00001$. Two-tailed Mann–Whitney test was used

many of these infections will occur in DENV-endemic regions, there is an urgency to understand the effect of pre-existing DENV immunity on ZIKV. A major question in the field is whether prior DENV immunity contributes to protection against or pathogenesis of ZIKV infection during pregnancy. To address this question, we adapted established mouse models of ZIKV infection during pregnancy that rely on acquired or genetic deficiencies of type I IFN signaling[10,11]. We challenged DENV-immune dams with ZIKV to model sequential DENV–ZIKV infection. In DENV-immune mice, we observed a reduction of ZIKV burden in maternal and fetal tissues, including the decidua/placenta, and an increase of fetal viability and growth compared to non-immune mice. Depletion or a genetic deficiency of CD8+ T cells abrogated this effect, demonstrating an essential role for CD8+ T cells in protection against ZIKV during pregnancy in the context of prior DENV immunity. Cross-reactive, polyfunctional CD8+ T cells during pregnancy may have the ability to overcome other pathogenic immune elements associated with prior DENV exposure, including ADE[42]. Indeed, we have previously reported that DENV-reactive CD8+ T cells can protect mice even under ADE conditions[32,36,49].

At an early time point after ZIKV infection of pregnant dams, CD8+ T cell depletion abrogated DENV-immune-mediated protection in maternal tissues and the maternal–fetal interface (i.e., decidua/placenta) despite having no effect on maternal viremia, suggesting that DENV-elicited memory CD8+ T cells preferentially exert their effects in tissues rather than in circulation. Accordingly, analysis of T cells in the maternal spleen

following ZIKV challenge revealed that the cross-reactive epitope-specific CD8+ T cell response was of greater magnitude and polyfunctionality in DENV-immune than non-immune dams. Thus, at early stage of ZIKV infection, prior DENV exposure elicited cross-reactive CD8+ T cells with greater functional activity compared to those expanded during primary ZIKV infection. Recent studies using blood samples from non-pregnant individuals have identified cross-reactive CD8+ T cells in humans[47,48]. One of these studies showed that DENV exposure prior to ZIKV infection influenced the magnitude and quality of the CD8+ T cell response[47], suggesting that prior DENV immunity may shape the anti-ZIKV CD8+ T cell response. A study with non-human primates suggested that prior DENV exposure may confer cross-protection against ZIKV infection[50], although a second study reported neither protective nor pathogenic effect of previous DENV exposure during subsequent ZIKV infection[51]. Notably, non-human primates in these studies were challenged 1–2 years following DENV exposure, as compared to our challenge of mice on day 30 after DENV priming. The transient nature of cross-protection mediated by DENV-primed CD8+ T cells against ZIKV infection during pregnancy is apparent based on our finding that challenge of mice on day 80 after priming did not afford cross-protection. This finding is consistent with the transient period of cross-protection, ranging from 1–2 weeks to 3 years, that is observed during human DENV infections[55–59]. Going forward, a detailed evaluation of the cross-reactive CD8+ T cell response is needed to understand the basis of the short-term nature. Knowledge of the mechanisms by which

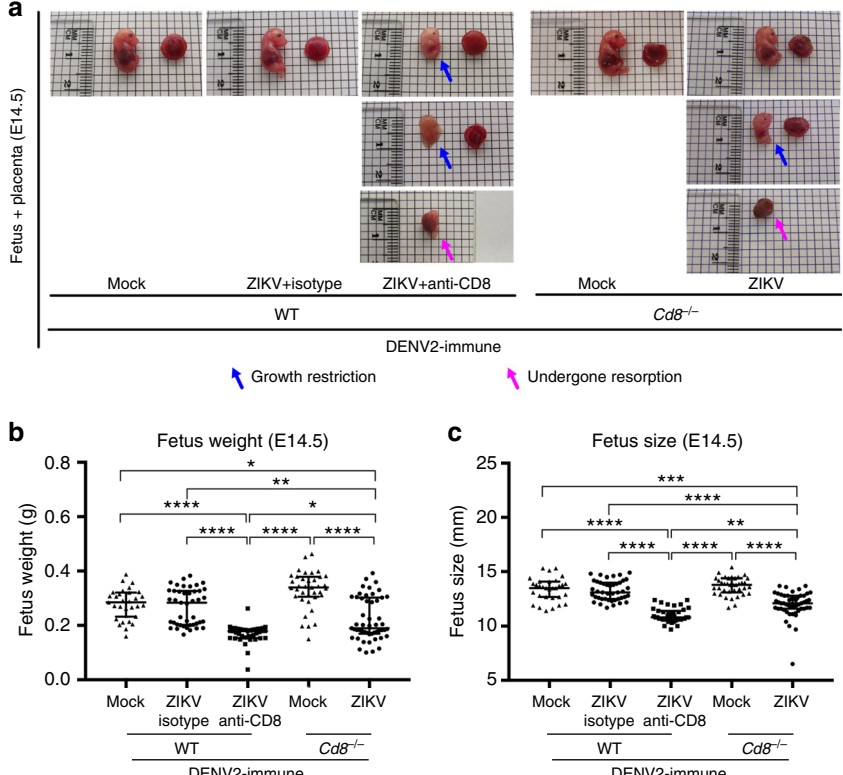

**Fig. 4** Phenotype of fetuses from DENV2-immune *Cd8^−/−* dams treated with Ifnar1-blocking Ab. DENV2-immune *Cd8^−/−* dams were generated 30 days after DENV2 priming, treated with Ifnar1-blocking Ab, and challenged with ZIKV at E7.5 as described in Fig. 2. Tissues were harvested 7 days after ZIKV infection at E14.5. **a** Representative images of fetuses and placentas are shown. *n* = 37 fetuses from five separate mothers (DENV2-immune-Mock), *n* = 51 fetuses from eight separate mothers (DENV2-immune-ZIKV). Numbers of WT fetuses and dams are described in Supplementary Fig. 3 (bottom panel). Blue and magenta arrows indicate the presence of fetal growth restriction and resorption, respectively. **b** Fetus weight and **c** size were recorded. Data were pooled from two independent experiments. Data were expressed as a median. *p < 0.05, **p < 0.01, ***p < 0.001, ****p < 0.0001. Kruskal–Wallis test was used for **b**, **c**

the cross-reactive CD8^+ T cell response wanes may enable development of strategies for enhancing the duration of cross-protection.

Epidemiologic studies have indicated (a) ~10–20% of people in northern Brazil are naive to DENV infection[62,63] and (b) an 11–13% risk of microcephaly to fetuses born to Brazilian mothers infected with ZIKV during pregnancy[63,64]. Indeed, these observations may be related, as maternal DENV immunity might prevent or reduce the impact of fetal ZIKV infection. However, human DENV–ZIKV cross-reactive immunological outcomes are likely based on a complex interplay of variables including but not limited to prior DENV infection history, gestational timing of ZIKV infection, route of ZIKV infection, maternal hormone status, HLA alleles, and additional determinants of individual T and B cell responses. We hypothesize that a robust maternal DENV–ZIKV cross-reactive CD8 T cell response could protect the fetus against ZIKV infection, but that fetal ZIKV infection results from mothers who are DENV naive or are DENV-immune but produce weak CD8 T cell responses with or without ADE conditions.

Consistent with the local effect of DENV-elicited CD8^+ T cells in each tissue, cross-reactive antigen-specific CD8^+ T cells also were detected in the decidua/placenta of DENV-immune mice. In agreement with CD8^+ T cells being one of the abundant cell types present in the decidua[60,65,66], the T cells detected in the decidua/placenta of DENV-immune dams were of maternal origin based on our allotype mating experiments. At present, the precise mechanism by which these cells balance immune tolerance of the

fetus and antiviral immunity at the maternal–fetal interface are presently unclear, but both virus-specific and fetal antigen-specific CD8^+ T cells have been detected in human and mouse decidua[5,60,65–69], and antigen-non-specific CD8^+ T cells that are pathogenic in a mouse model of LPS-induced intrauterine inflammation[70] also may be important. The decidual CD8^+ T cells in humans are primarily of effector memory phenotype and express reduced levels of granzyme B compared to peripheral blood CD8^+ T cells[65,69]. Consistent with this observation, cross-reactive antigen-specific CD8^+ T cells in the decidua/placenta of DENV-immune dams with ZIKV infection were effector memory, the majority of which had polyfunctional capacity, as defined by granzyme B or both IFNγ and TNF expression. Notably, despite the reported epigenetic silencing of chemokine genes in the decidua, which would limit T cell access during pregnancy[61], more antigen-specific CD8^+ T cells were present in the decidua/placenta of DENV-immune than non-immune mice. Future studies are needed to determine the mechanisms by which these T cells were recruited or activated locally in the decidua.

Immune responses during pregnancy are complex and remain poorly understood. Little is known about the immune response to ZIKV infection during pregnancy, except for a recent study that reported a decreased frequency of granzyme B expressing total CD8^+ T cells in pregnant dams compared to non-pregnant mice[71], suggesting that the anti-ZIKV T cell response quantity or quality may be reduced during pregnancy. This published study and our data set the framework for comparing antigen-specific CD8^+ T cell responses in pregnant and non-pregnant mice with

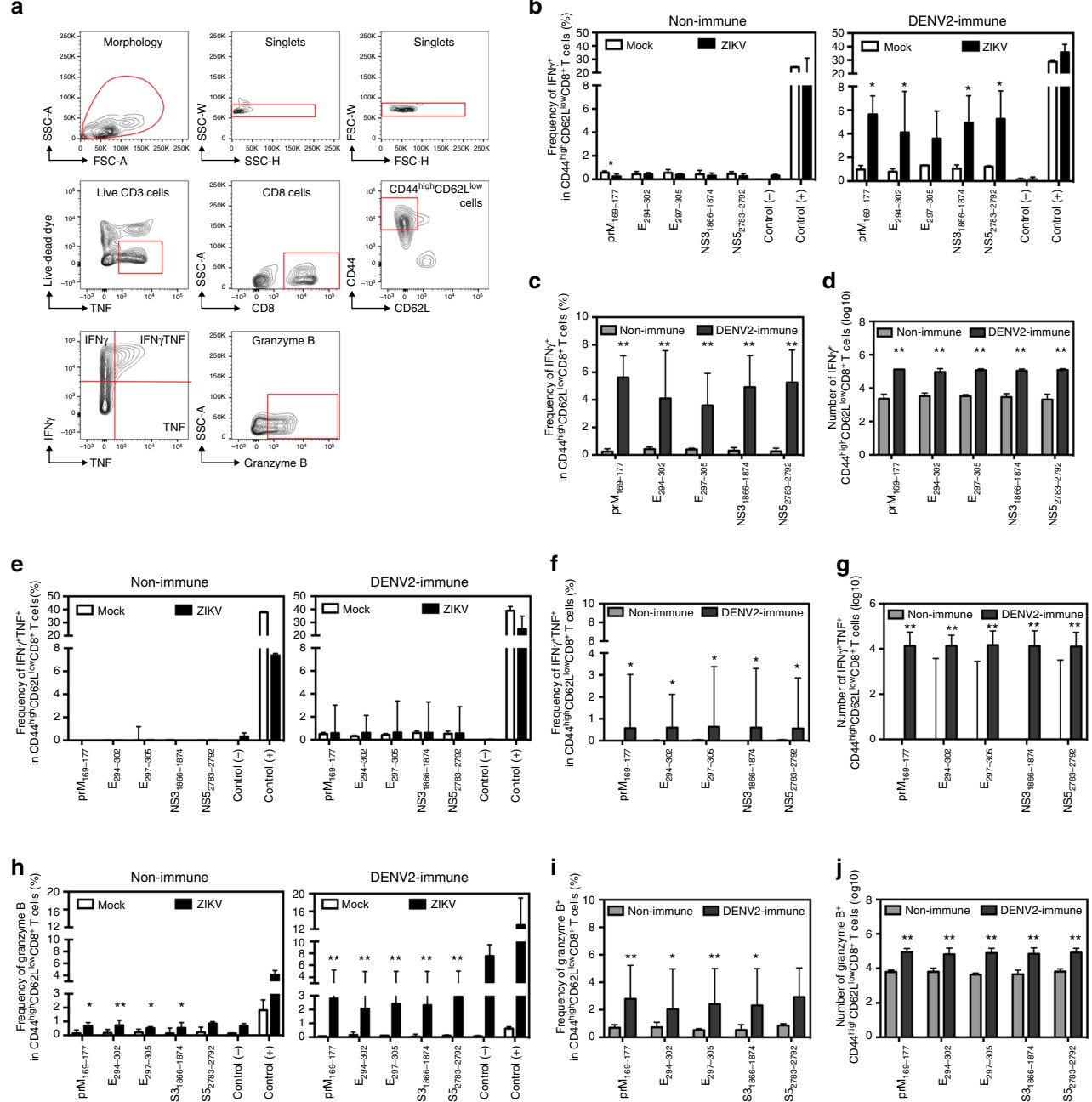

**Fig. 5** Cross-reactive epitope-specific CD8$^+$ T cell response in spleens from Ifnar1-blocking Ab-treated WT dams on day 3 after ZIKV infection. Non-immune or DENV2-immune WT dams with transient Ifnar1 blockade were challenged with ZIKV at E7.5 or injected 10% FBS–PBS (MOCK) as described in Fig. 2. Three days after ZIKV infection (E10.5), mice were killed, and spleens were processed for ICS analysis. **a** The gating strategy is represented. **b–d** The percentages (**b**, **c**) and numbers (**d**) of CD44$^{high}$CD62L$^{low}$ CD8$^+$ T cells producing IFNγ are shown. **e–g** The percentages (**e**, **f**) and numbers (**g**) of CD44$^{high}$CD62L$^{low}$ CD8$^+$ T cells producing both IFNγ and TNF are shown. **h–j** The percentages (**h**, **i**) and numbers (**j**) of CD44$^{high}$CD62L$^{low}$ CD8$^+$ T cells expressing granzyme B are shown. The following numbers of dams were used: Non-immune ($n = 6$ MOCK and $n = 6$ ZIKV) and DENV2-immune ($n = 6$ MOCK and $n = 4$ ZIKV). Data represent a pool of three independent experiments, and are expressed as median with interquartile range. *$p < 0.05$, **$p < 0.01$. Two-tailed Mann–Whitney test was used to compare MOCK versus ZIKV-infected or non-immune versus DENV2-immune mice for each stimulation condition in **b–j**

ZIKV infection. Given that gestational stage influences the susceptibility of ZIKV infection in the placenta and fetus[72], it will be important also to evaluate the temporal component of the anti-ZIKV T cell response through the different stages of pregnancy.

We previously demonstrated that CD8$^+$ T cells are necessary and sufficient to protect against systemic ZIKV challenge in both naive and DENV-immune non-pregnant mice[39,46,49]. Here, we observed a similar requirement for CD8$^+$ T cells in protection

against ZIKV in the context of pregnancy and prior DENV exposure. We also observed a partially protective role for CD4$^+$ T cells in this study, suggesting that CD4$^+$ T cell-mediated-help may shape an optimal cross-reactive CD8$^+$ T cell response during ZIKV infection of DENV-immune pregnant females. Alternatively, CD4$^+$ T cells may exert their effect by regulating humoral immunity or CD4$^+$ regulatory T cells could minimize pathology at the maternal–fetal interface.

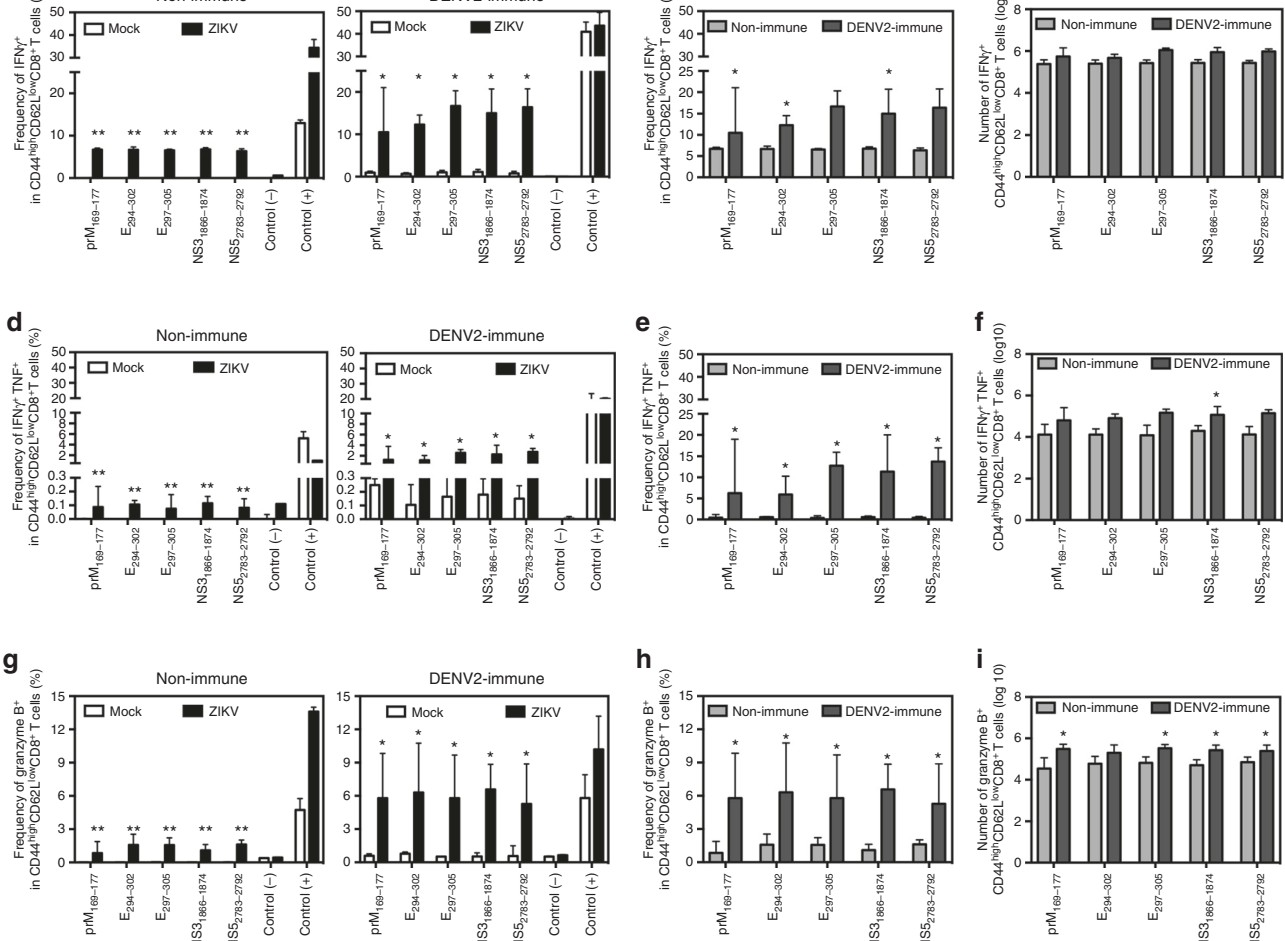

**Fig. 6** Cross-reactive epitope-specific CD8+ T cell response in spleens from Ifnar1-blocking Ab-treated WT dams on day 7 after ZIKV infection. Non-immune or DENV2-immune WT dams with transient Ifnar1 blockade were challenged with ZIKV at E7.5 or injected 10% FBS–PBS (MOCK) as described in Fig. 2. Seven days after ZIKV infection (E14.5), mice were killed, and spleens were processed for ICS analysis. **a–c** The percentages (**a**, **b**) and numbers (**c**) of CD44highCD62Llow CD8+ T cells producing IFNγ are shown. **d–f** The percentages (**d**, **e**) and numbers (**f**) of CD44highCD62Llow CD8+ T cells producing both IFNγ and TNF are represented. **g–i** The percentages (**g**, **h**) and numbers (**i**) of CD44highCD62Llow CD8+ T cells expressing granzyme B are shown. The following numbers of dams were used: Non-immune ($n = 6$ MOCK and $n = 5$ ZIKV) and DENV2-immune ($n = 4$ MOCK and $n = 5$ ZIKV). Data were pooled from two independent experiments and are expressed as median with interquartile range. *$p < 0.05$, **$p < 0.01$. Two-tailed Mann–Whitney test was used to compare MOCK versus ZIKV-infected or non-immune versus DENV2-immune mice for each stimulation condition in **a–i**

Our study also raises key issues of epidemiologic relevance particularly in terms of the T cell response to ZIKV infection in individuals with previous DENV exposure. The following questions are unanswered: Is the cross-protection against ZIKV during pregnancy dependent on prior infecting DENV serotype? In secondary DENV infection cases, how does the sequence of infection with different DENV serotypes influence subsequent ZIKV infection during pregnancy? What is the duration of cross-protection against ZIKV infection during pregnancy? Answers to these questions will likely provide insight into the natural history of ZIKV infection in pregnancy in DENV-endemic areas and also inform the feasibility of developing combination vaccines that protect against both ZIKV and DENV. As CD8+ T cell responses induced by a tetravalent live-attenuated DENV vaccine also cross-reacted with ZIKV epitopes[47], further boosting of T cell responses could confer protection against ZIKV infection in pregnancy. Moreover, ZIKV vaccines that are designed to induce optimal T cell responses in addition to Abs may be more effective than those that focus solely on Ab responses for protection against ZIKV during pregnancy.

## Methods

**Ethics statement.** This study was performed following the guidelines of the Institutional Animal Care and Use Committee under protocol #AP028-SS1-0615. La Jolla Institute for Allergy and Immunology (LJI) has established an animal care and use program in compliance with The Public Health Service Policy on the Humane Care and Use of Laboratory Animals, and maintains an animal welfare assurance with the Office of Laboratory Animal Welfare (OLAW). The animal care and use program is guided by the US Government Principles for the Utilization and Care of Vertebrate Animals Used in Testing, Research and Training and by the 8th edition of the Guide for the Care and Use of Laboratory Animals. As such, all research involving animals are reviewed and approved by the IACUC in accordance with The PHS policy on the Humane Care and Use of Animals and the 8th edition of The Guide. In addition, LJI's animal care and use program is accredited by AAALAC International. Inoculations were performed under isoflurane inhalation, and all efforts were made to minimize pain.

**Viruses.** ZIKV Asian lineage strain FSS13025 (Cambodia, 2010) was obtained from the World Reference Center for Emerging Viruses and Arboviruses (Galveston, TX). The mouse-adapted DENV2 strain S221 is a biological clone derived from DENV2 D2S10 [33,73]. ZIKV and DENV2 were propagated in C6/36 *Aedes albopictus* cells (ATCC® CRL-1660™), and viral titers were measured by focus forming assay (FFA)[39] with baby hamster kidney (BHK)-21 cells were purchased from the American Type Culture Collection (ATCC) or by qRT-PCR[74].

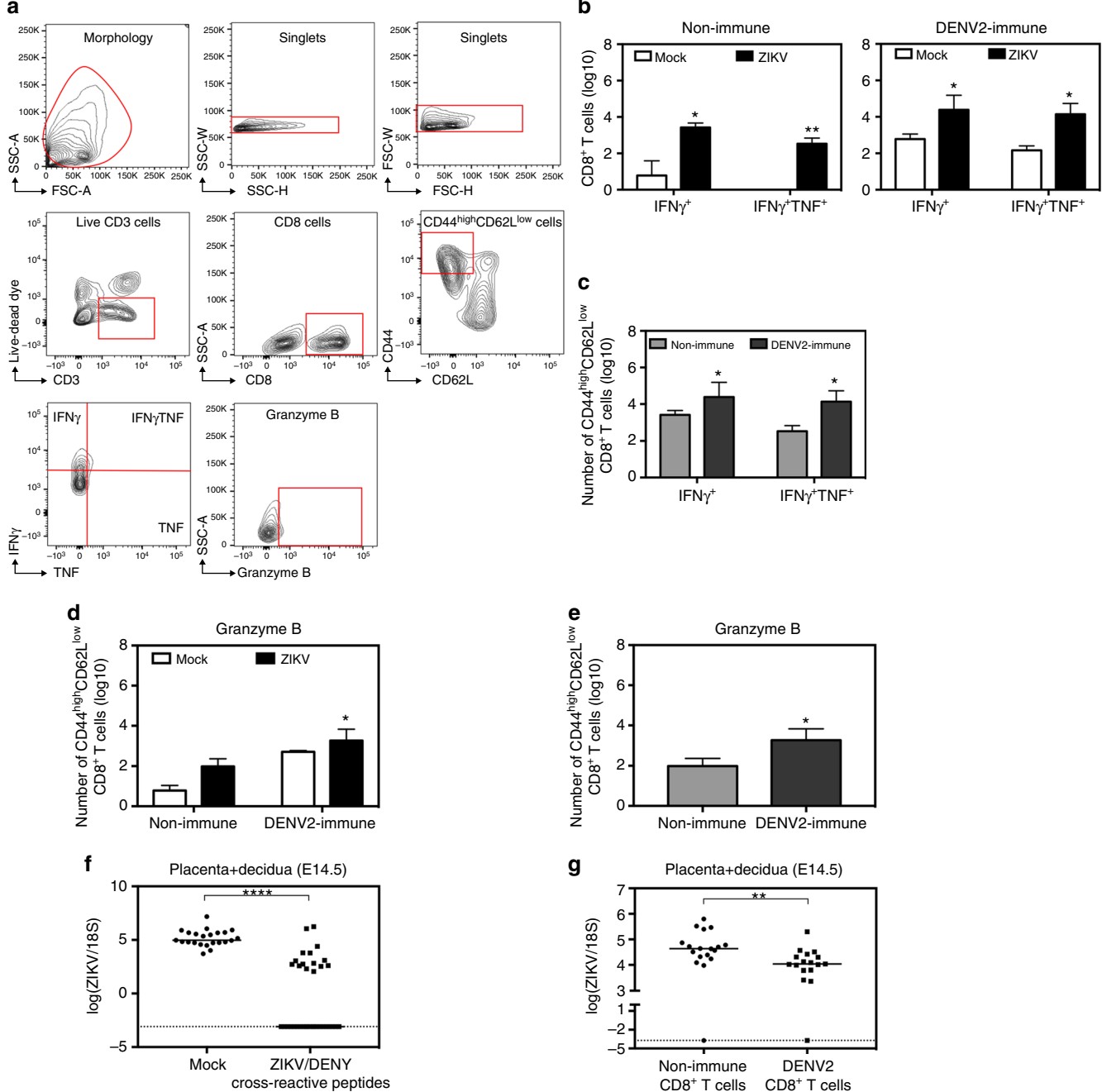

**Fig. 7** Cross-reactive epitope-specific CD8+ T cell response in placenta with decidua of non-immune or DENV2-immune Ifnar1 mAb-treated WT dams 7 days after ZIKV infection. **a–e** Non-immune or DENV2-immune WT dams with Ifnar1 blockade were challenged with ZIKV at E7.5 as described in Fig. 2. Placentas with decidua were harvested 7 days post infection at E14.5. Isolated cells were stimulated with a pool of five cross-reactive peptides for ICS. **a** The gating strategy used to analyze the ICS data is illustrated. **b–e** The number of CD44highCD62Llow CD8+ T cells expressing IFNγ or both IFNγ and TNF (**b**, **c**) or granzyme B (**d**, **e**) are shown. A total of ten non-immune (n = 6 MOCK and n = 4 ZIKV) and nine DENV2-immune (n = 4 MOCK and n = 5 ZIKV) dams were analyzed. **f** Virgin female WT mice were immunized subcutaneously with a mixture of five cross-reactive peptides (ZIKV/DENV peptides + CFA) or with DMSO (Mock + CFA). Two weeks later, mice were boosted with IFA adjuvant and mated with WT sires. Fourteen days after the boost, on E7.5, dams were challenged with 10^4 FFU of ZIKV FSS13025. ZIKV RNA levels were quantified in the placenta/decidua of ZIKV/DENV peptide-immunized (n = 36 fetuses from five dams) and mock-immunized (n = 23 fetuses from three dams) groups on day 7 post infection (E14.5). **g** A total of 1.5 × 10^7 CD8+ T cells isolated from spleens of DENV2-immune or non-immune mice were transferred into recipient pregnant dams on E6.5 1 day prior to challenge with 10^4 FFU of ZIKV FSS13025. ZIKV RNA levels in the placenta with decidua at E14.5 in mice that received DENV2-primed (n = 17 fetuses) or non-immune (n = 18 fetuses) CD8+ T cell were measured by qRT-PCR. Data for adoptive transfer were obtained from two separate mothers for each group. ICS data were expressed as median with interquartile range and pooled from two independent experiments. Viral burden data are expressed as a median. *p < 0.05, **p < 0.01, ****p < 0.0001. Two-tailed Mann–Whitney test was used to compare MOCK versus ZIKV and non-immune versus DENV2-immune dams, Mock versus ZIKV/DENV cross-reactive peptides and non-immune CD8+ T cells versus DENV2-CD8+ T cells

**Mouse experiments and virus infections**. $Ifnar1^{-/-}$ and wild-type (WT) congenic C57BL/6J mice were bred in a specific pathogen-free facility at La Jolla Institute for Allergy & Immunology, or WT mice (males, #000664 and CD45.1 #002014) also were purchased from Jackson Laboratories. $Cd8^{-/-}$ C57BL/6 mice were purchased (The Jackson Laboratory, 002665) and bred at Washington University School of Medicine, St. Louis. Two models of ZIKV pregnancy infection were used: (1) $Ifnar1^{-/-}$ females crossed to WT males and (2) WT females crossed to WT males. The type I IFN receptor (Ifnar1) signaling in WT females was transiently blocked via treatment with an Ifnar1-blocking monoclonal Ab (mAb), as described below.

To establish DENV immunity, 5-week-old $Ifnar1^{-/-}$ female mice were inoculated with $1 \times 10^3$ FFU of DENV2 via IP route or WT female mice were inoculated with $1 \times 10^4$ FFU of DENV2 via RO route. At 8–10 weeks of age, female mice were mated. Pregnancy was determined by the presence of a vaginal plug in the morning, and embryonic development was estimated as gestational age E0.5. Pregnant female mice were separated from male mice after plug detection. At E7.5, the females were inoculated with $1 \times 10^4$ FFU of ZIKV in 20 or 200 µL of PBS with 10% FBS or mock-infected with 20 or 200 µL of PBS with 10% FBS via IF or RO route, respectively. Mice were killed at E10.5 or E14.5 depending on the experimental design. Viral burden in the maternal tissues (serum, brain, and spleen), placenta, and fetal tissues (head and body) were quantified. Fetus weight and size were measured using an analytical balance (Catalog number: S94790A, Fisher Scientific) and Digital Caliper (Model number: 700-113-10, Mitutoyo), respectively.

**Ifnar1 blocking and T cell-depleting antibodies**. All antibodies (Abs) were purchased from BioXCell (USA). For Ifnar1 blocking, WT mice were treated with 2 mg Ifnar1-blocking mAb MAR1-5A3 via IP route 1 day before infection with ZIKV or DENV2 (WT mice only). To deplete CD8+ T cells, mice were injected with either anti-mouse CD8 mAb (300 µg/mouse, rat IgG2b, clone 2.43) or isotype control mAb (300 µg/mouse, rat IgG2b, clone LTF-2) via RO route on days 3 and 1 prior to infection and every 2 days after ZIKV infection. The same protocol was used for CD4+ T cell depletion (300 µg/mouse, clone GK1.5) or both CD4+ and CD8+ T cell depletion. All mice were monitored for CD8+ T cell depletion in tissues after treatment using flow cytometry.

**qRT-PCR analysis of viral burdens**. For RNA extraction, organs were collected in RNA later (Ambion) and stored at 4 °C. Tissues were homogenized in BME/RLT buffer for 3 min using Tissue lyser II (QIAGEN) and then centrifuged for 1 min at $6010 \times g$. RNA from tissue samples and serum obtained from ZIKV-infected mice were extracted using the RNeasy Mini Kit (tissues) and Viral RNA Mini Kit (serum) (QIAGEN). All RNA samples were stored at −80 °C. For quantification of viral RNA, real-time qRT-PCR was performed using the qScript One-Step qRT-PCR Kit (Quanta BioSciences) at the CFX96 TouchTM real-time PCR detection system (Bio-Rad CFX Manager 3.1). A published primer set[74] was used to detect ZIKV RNA: Fwd, 5′-TTGGTCATGATACTGCTGATTGC-3′; Rev, 5′-CCTTCCACAAAGTCCCTATTGC-3′ and Probe, 5′-6-FAM-CGGCATA-CAGCATCAGGTGCATAGGAG-Tamra-Q-3′. Cycling conditions were set as follows: 45 °C for 15 min, 95 °C for 15 min, followed by 50 cycles of 95 °C for 15 s and 60 °C for 15 s and a final extension of 72 °C for 30 min. Viral RNA concentration was determined based on an internal standard curve composed of serial dilutions of an in vitro transcribed RNA based on ZIKV strain FSS13025.

**Peptide synthesis**. Peptides were purchased from Synthetic Biomolecules (A&A Labs). The 9- and 10-mer peptides used for flow cytometry were synthesized and purified by reverse-phase HPLC up to ≥95% purity. Peptides were stored at −20 °C after being dissolved in DMSO and aliquoted into small quantities to avoid freeze-thaw damage. The sequence and characteristics of all peptides used have been published[49].

**Cell isolation and flow cytometric analyses**. For each pregnant mouse, placentas were harvested in 10% FBS/RPMI and pooled before processing as follows. Briefly, placentas without separation of maternal decidua were cut into small pieces and treated with 1 mg/ml of type I collagenase (Worthington) for 60 min at 37 °C. After incubation, placentas were mechanically dissociated, filtered through over a 70-µm cell strainer and the pellet was resuspended in 44% Percoll (GE Healthcare). Another layer of 67% Percoll was placed underneath before centrifugation at $376 \times g$ at room temperature for 20 min. The cell layer suspension was isolated between the different densities of Percoll and washed three times with PBS. Cells were counted after erythrocyte lysis using a cell counter (Vi-cell XR 2.04, Beckman Coulter).

For ICS, isolated splenocytes from all mice were plated as $2 \times 10^6$ splenocytes/well in 96-well U-bottom plates. Cells were stimulated with 1 ug of individual ZIKV peptides for 6 h in the presence of Brefeldin A (GolgiPlug; BD Biosciences) during the last 4 h. Cells from placenta/maternal decidua were plated and stimulated with a mixture of all five ZIKV peptides following the same conditions as splenocytes. Positive controls using a cell stimulation cocktail (commercial PMA-Ionomycin-500×, eBiosciences) and negative controls (10% FBS/RPMI) were added in each plate. Cells were washed after stimulation and labeled with viability dye efluor 455 UV (1000×, Invitrogen) in PBS. All cells were stained at 1:200 dilution with anti-

CD3 PerCpCy 5.5 (Clone 145-2C11), anti-CD8 BV510 (clone 53–67), anti-CD44 BV785 (clone IM7), anti-CD62L APC eFluor 780 (clone Mel-14), followed by fixation and permeabilization using the BD Cytofix/Cytoperm kit and then staining with a combination of anti-IFNγ FITC (clone XMG 1.2), anti-TNF Alexa Fluor 700 (clone MP6-XT22), and granzyme B PE-Cy7 (clone NGZB). For staining of CD45 allelic markers, cells were incubated with anti-CD45.1 APC (clone A20) and anti-CD45.2 FITC (clone 104). Samples were acquired on LSR-Fortessa (BD Biosciences) and analyzed using FlowJo software X 10.0.7 (Tree Star).

**Peptide immunization**. Five-week-old C57BL/6 mice were immunized with 50 µg of a mixture of five cross-reactive DENV/ZIKV peptides (PrM$_{169-177}$, E$_{294-302}$, E$_{297-305}$, NS3$_{1866-1874}$, and NS5$_{2783-2792}$) or DMSO with complete Freund's adjuvant (CFA) (v/v). Two weeks later, mice were boosted with the same mixture of peptides or DMSO + incomplete Freund's adjuvant (IFA). Fourteen days after the boost, mice were challenged with $10^4$ FFU of ZIKV FSS13025, and this time point corresponds to E7.5 of pregnancy. Tissues were collected from pregnant dams on day 7 post infection with ZIKV (E14.5). ZIKV RNA levels were quantified via qRT-PCR.

**Adoptive transfer of T cells**. Eight- to ten-week-old C57BL/6 mice were treated with Ifnar1-blocking Ab 1 day before challenge with $10^4$ FFU of DENV2 strain S221 or 10% FBS/PBS to generate DENV-immune or non-immune T cell donor mice, respectively. DENV-immune or non-immune CD8+ T cells were isolated 30 days after DENV2 infection. Briefly, spleens were harvested and CD8+ T cells were positively isolated using antibody-coated microbeads from CD8 T Cell Isolation Kit (Miltenyi Biotec). The purity of the isolated population was checked by flow cytometry (>90% of purity). A total of $1.5 \times 10^7$ CD8+ T cells (DENV2-immune or non-immune CD8 T cells) were intravenously transferred to pregnant mice treated with 2 mg of Ifnar1-blocking Ab at E6.5. One day later, mice were infected with $10^4$ FFU of ZIKV FSS13025. ZIKV RNA levels were quantified in the placenta/decidua at day 7 after ZIKV infection (E14.5).

**Statistical analysis**. All data were analyzed with Prism software, version 7.0 (GraphPad Software). For morphological measurements and viral burden data, a Kruskal–Wallis test was used to compare more than two groups. Viral burden data and ICS data were analyzed using a two-tailed Mann–Whitney test to compare two groups. Percentages of infection in placenta with decidua and fetal tissues were assessed via two-sided Fisher's exact test. For morphological measurements and viral burden, data were expressed as a median. ICS data were shown as an inter-quartile range. $p < 0.05$ was considered a significant difference.

**Data availability**. The authors declare that all data supporting the findings of this study are available within the paper and its supplementary information files.

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

## Acknowledgements

This research was funded by NIH grants (R01 AI116813 and R21 NS100477 to S.S. and P01 AI106695, R01 AI073755, and R01 AI127828 to M.S.D.) and the Chiba-UCSD Center for Mucosal Immunology, Allergy and Vaccine Development.

## Author contributions

J.A.R.-N., A.E.N., and S.S. designed the project. J.A.R.-N. and A.E.N. designed, performed, and analyzed experiments. K.M.V., A.-T.H., Y.-T.W., A.-V.T.N., R.S., and A.M. performed the experiments. J.A.R.-N., A.E.N., Y.-T.W., K.K. and S.S. interpreted the data and wrote the manuscript. M.S.D. and S.S. edited the manuscript.

## Additional information

**Competing interests:** M.S.D. is a consultant for Inbios and Sanofi, and is on the Scientific Advisory Board of Moderna. The remaining authors declare no competing interests.



