## [Peer Review File · Nature Communications]

Reviewers' comments:

Reviewer #1 (Remarks to the Author):

Zika Virus is a flavivirus and infections in Micronesia have been associated with Guillain Barre Syndrome. In children born to some infected pregnant women in South and Central America, microcephaly and other birth defects have been associated with ZIKV infection. There is significant interest in understanding what cell types are susceptible to ZIKV infection at the maternal-fetal interface and developing models that mimic these interactions. Recent publications suggest cross-reactivity between DENV (a related flavivirus that co-circulates in similar regions) and ZIKV at the antibody and T cell level. In this study, the authors evaluated (i) how prior DENV exposure influences maternal and fetal outcome of ZIKV infection in pregnancy and (ii) the contribution of CD8+ T cells to protection from or pathogenesis of ZIKV infection during pregnancy. There are several concerns with the model and interpretation of the findings that need to be addressed.

1. The experiments were carried out in mice with substantially altered immune systems (Ifnar1-/- or ab blockade of Type 1 IFN in wildtype mice) to achieve dengue and zika infection. In a recent report by Yockey et al. The authors demonstrate that type I interferon signaling is pathogenic in a mouse model of congenital ZIKV infection, resulting in abnormal placental architecture, intrauterine growth restriction, and fetal demise. This paper challenges the use of the current model in the manuscript (Dengue Virus CD8+ T Cells Cross-protect against Zika Virus During Pregnancy).

Yockey et al. Type I interferons instigate fetal demise after Zika virus infection. *Science Immunology* 05 Jan 2018: Vol. 3, Issue 19, eaao1680. PMID: 29305462.

2. In this paper, the authors use the retro-orbital route for zika infection. In addition, Abs were also administered via this route. It is puzzling why the authors did not use a more physiological route of infection. The route of infection is likely to affect the outcome in congenital infections. There are models published using the intravaginal or subcutaneous infection.

Yockey LJ et al. Vaginal Exposure to Zika Virus during Pregnancy Leads to Fetal Brain Infection. *Cell*. 2016; 166(5):1247±56 e4. PMID: 27565347.

Miner, J.J. et al. (2016) Zika virus infection during pregnancy in mice causes placental damage and fetal demise. *Cell* 165, 1081–1091. PMID: 27180225

3. The authors depleted CD8 T cells from mice with prior exposure to dengue (>30 days post DENV infection). Do the authors know how long the Dengue virus CD8 T cell cross-protection to Zika could be maintained? DENV immune individuals exposed to ZIKV are likely to be exposed to DENV several years prior. The authors do state this is a limitation but it is a major concern whether recent immunity to DENV truly reflects how humoral and cellular subsets respond to subsequent ZIKV infection. A recent report *Emerg Infect Dis*. 2017 May;23(5):773-781. supports findings of a lack of durable cross-reactive humoral immunity between DENV and ZIKV.

4. The authors state in their discussion "DENV-elicited memory CD8+ T cells preferentially exert their effects in tissues rather than in circulation". The paper by Yockey et al suggests that it is Type 1 IFN that mediates effects in their ZIKV congenital model. How do Dengue virus CD8 T cells mediate protection? The authors do stain for intracellular cytokines (IFN- γ , TNF) and a cytolytic molecule (GrzB) in CD8 T cells from the spleen and decidua. Is this protection mediated by receptor, cytokines or other effector molecules? The authors should also consider adoptive transfer studies with one of the many cross-reactive peptides they found to stimulate CD8+ T cells in the spleen and decidua.

5. The ANOVA test with Tukey's multiple comparison tests are commonly used for a normal (Gaussian) distribution. Did the author's test for normality of the data? Or did they transform the data? The authors should test for normality of data distribution (for example using histograms, stem-and-leaf plots, or normal probability plots or hypothesis tests, such as the Kolmogorov-Smirnov and Shapiro-Wilk tests) to know whether to use a parametric or non-parametric test. Data analyzed by nonparametric statistics are commonly presented as the median along with an appropriate range (minimum and maximum values, upper and lower quartiles or quintiles, etc.) and not use mean \pm SEM. The following articles are useful for statistics:

Martin Krzywinski & Naomi Altman. Points of significance: Nonparametric tests. *Nature Methods* 11, 467–468 (2014). PMID: 24820360.
Cara H. Olsen. Statistics in Infection and Immunity Revisited. *Infect Immun.* 2014 Mar; 82(3): 916–920. PMID: PMC3957975

Reviewer #2 (Remarks to the Author):

The major claim of the paper is that cross-reactive CD8 cells (against Dengue) may protect against Zika virus. However, the main premise is not fully tested. Maternal CD8 cells (placental infiltrates) are known to be associated with adverse perinatal outcomes as in cases of intrauterine inflammation (e.g., Lei et al., 2017). Therefore, the title and the premise of the manuscript may deal only with viral transmission but not necessarily the adverse perinatal outcomes associated with CD8 cells. As is the manuscript is well-written but lacks translational potential.

Reviewers' comments:

Reviewer #1 (Remarks to the Author):

Zika Virus is a flavivirus and infections in Micronesia have been associated with Guillain Barre Syndrome. In children born to some infected pregnant women in South and Central America, microcephaly and other birth defects have been associated with ZIKV infection. There is significant interest in understanding what cell types are susceptible to ZIKV infection at the maternal-fetal interface and developing models that mimic these interactions. Recent publications suggest cross-reactivity between DENV (a related flavivirus that co-circulates in similar regions) and ZIKV at the antibody and T cell level. In this study, the authors evaluated (i) how prior DENV exposure influences maternal and fetal outcome of ZIKV infection in pregnancy and (ii) the contribution of CD8+ T cells to protection from or pathogenesis of ZIKV infection during pregnancy. There are several concerns with the model and interpretation of the findings that need to be addressed.

We would like to thank the reviewer for the positive comments about the importance of our study and providing suggestions for improving the manuscript.

1. The experiments were carried out in mice with substantially altered immune systems (Ifnar1^{-/-} or ab blockade of Type I IFN in wildtype mice) to achieve dengue and Zika infection. In a recent report by Yockey et al. The authors demonstrate that type I interferon signaling is pathogenic in a mouse model of congenital ZIKV infection, resulting in abnormal placental architecture, intrauterine growth restriction, and fetal demise. This paper challenges the use of the current model in the manuscript (Dengue Virus CD8+ T Cells Cross-protect against Zika Virus During Pregnancy).

Yockey et al. Type I interferons instigate fetal demise after Zika virus infection. Science Immunology 05 Jan 2018: Vol. 3, Issue 19, eaao1680. PMID: 29305462.

As ZIKV cannot replicate efficiently in mice with intact type I interferon (IFN) system, we and other investigators have used Ifnar1^{-/-} dams mated with wild-type sires to generate Ifnar1^{+/-} heterozygous fetuses that have functional type I IFN signaling^{1,2}. In contrast, Yockey et al. mated Ifnar1^{-/-} females with Ifnar1^{+/-} males to reveal that Ifnar1^{+/-} but not Ifnar1^{-/-} fetuses were resorbed upon maternal ZIKV infection; Ifnar1^{-/-} fetuses continued their development in ZIKV-infected pregnant mothers³. Our study focused on investigating the role of CD8 T cells in protecting against pathogenesis (rather than causing it) during maternal ZIKV infection. Although type I IFN indeed could be pathogenic as Yockey et al suggest, DENV-primed CD8 T cells still played an important role in protecting against ZIKV. Moreover, in addition to the genetic KO model, we used WT females that were mated with WT males and transiently pre-treated with blocking anti-IFNAR1 antibody one day prior to infection. In both models, we still observed CD8 T cell-mediated protection against fetal growth restriction or demise.

2. In this paper, the authors use the retro-orbital route for zika infection. In addition, Abs were also administered via this route. It is puzzling why the authors did not use a more physiological route of infection. The route of infection is likely to affect the outcome in congenital infections. There are

models published using the intravaginal or subcutaneous infection.

Yockey LJ et al. Vaginal Exposure to Zika Virus during Pregnancy Leads to Fetal Brain Infection. Cell. 2016; 166(5):1247±56 e4. PMID: 27565347.

Miner, J.J. et al. (2016) Zika virus infection during pregnancy in mice causes placental damage and fetal demise. Cell 165, 1081–1091. PMID: 27180225

We determined whether other routes of ZIKV inoculation could mediate fetal resorption. Ifnar1^{-/-} females crossed with WT males were challenged via the subcutaneous route with 10⁴ FFU (in the footpad) of ZIKV FSS13025 on E7.5. Following ZIKV challenge, all fetuses were resorbed at E14.5, but with a slightly less fetus damage (in terms of fetus weight and size) compared with a retro-orbital (R.O.) infection route (Fig S1). This is consistent with many published DENV studies in mice⁴⁻⁷ and a DENV study in monkeys⁸ showing that the intravenous (I.V.) route of DENV inoculation produces a relevant and uniformly consistent phenotype. In recent ZIKV studies, I.V. route has also been used to evaluate viral replication and immune responses after ZIKV challenge in rhesus macaques⁹ and to demonstrate fetal damage in immunocompetent pregnant mice ZIKV infection¹⁰. In our study, Abs were administered retro-orbitally instead of intraperitoneally in order to avoid damage in the uterus during multiple injections.

Based on reviewer's comment, we have added the information about our using the intravenous and subcutaneous routes of virus inoculation in our manuscript (page 5, lines 93-99).

3. The authors depleted CD8 T cells from mice with prior exposure to dengue (>30 days post DENV infection). Do the authors know how long the Dengue virus CD8 T cell cross-protection to Zika could be maintained? DENV immune individuals exposed to ZIKV are likely to be exposed to DENV several years prior. The authors do state this is a limitation but it is a major concern whether recent immunity to DENV truly reflects how humoral and cellular subsets respond to subsequent ZIKV infection. A recent report Emerg Infect Dis. 2017 May;23(5):773-781. supports findings of a lack of durable cross-reactive humoral immunity between DENV and ZIKV.

We agree with this comment. To address the reviewer's question, we evaluated whether cross-protection was still observed in dams primed with DENV for 80 days prior to ZIKV challenge. Based on the levels of ZIKV RNA measured by qRT-PCR, cross-protection was detected in the maternal spleen; however, no differences were observed between DENV-immune and non-immune dams in terms of the levels of ZIKV RNA in placenta/decidua and the fetal weight and size (Fig. S7). This short duration of cross-protection is consistent with published DENV studies reporting a range of 1-2 weeks to 3 years as the length of cross-protection¹¹⁻¹⁵. Our study provides a foundation for future studies to explore mechanisms that can be manipulated for enhancing the duration of cross-protective immunity. Augmenting the period of cross-protection is a critical issue for designing a safe and effective ZIKV vaccine. Information about these new results can be found in our revised manuscript: (page 7, line 175-182; page 10 and 11, lines 267-273).

4. The authors state in their discussion "DENV-elicited memory CD8+ T cells preferentially exert their effects in tissues rather than in circulation". The paper by Yockey et al suggests that it is Type 1 IFN that mediates effects in their ZIKV congenital model. How do Dengue virus CD8 T cells mediate

protection? The authors do stain for intracellular cytokines (IFN- γ , TNF) and a cytolytic molecule (GrzB) in CD8 T cells from the spleen and decidua. Is this protection mediated by receptor, cytokines or other effector molecules? The authors should also consider adoptive transfer studies with one of the many cross-reactive peptides they found to stimulate CD8⁺ T cells in the spleen and decidua.

We appreciate the reviewer's suggestions. Accordingly, we took multiple approaches to further understand the nature of the cross-protective CD8 T cell response. Per the reviewer's suggestion, we first performed peptide-immunization experiments. Mice were immunized with a mixture of ZIKV/DENV cross-reactive peptides and challenged at E7.5. At day 7 post-infection, ZIKV RNA levels in the placenta/decidua from peptide immunized mice were significantly decreased compared to mock-immunized mice (Fig. 6F). Second, we performed adoptive transfer studies in which DENV2-primed or naïve CD8 T cells were adoptively transferred into naïve pregnant recipients at E6.5 prior to challenge with ZIKV at E7.5. ZIKV-RNA levels were significantly decreased in mice transferred with DENV2-primed CD8 T cells compared to naïve CD8⁺ T cell recipients (Fig. 6G). Collectively, the peptide immunization and adoptive transfer studies demonstrated that cross-reactive CD8 T cells by themselves were sufficient to limit ZIKV infection in the placenta/decidua. These new studies have been described both in the Materials and Methods (page 22-23, line 600-615) and Results sections (page 9, lines 226-237).

Third, to validate our results obtained from CD8 T cell depletion studies, we used gene-deficient mice lacking CD8 T cells. Congenic Cd8^{-/-} and WT C57BL/6 DENV2-immune dams were treated with Ifnar1 blocking antibody, followed by ZIKV challenge at E7.5 and harvest at E14.5. In Cd8^{-/-} mice, the fetuses showed growth restriction and underwent resorption, in agreement with results obtained from experiments using CD8 T cell-depleted WT mice (Fig S6). These results with Cd8^{-/-} mice have been added in the manuscript (page 7, lines 169-174).

Finally, to demonstrate the maternal origin of cross-protective T cells in the placenta with decidua, we mated CD45.2⁺ DENV2-immune WT females with CD45.1⁺ WT males and performed intracellular cytokine staining of cells isolated from the placenta/decidua after restimulation with cross-reactive peptides. We observed that >95% of leukocytes at the maternal-fetal interface were CD45.2⁺, indicating maternal origin of cross-reactive CD8 T cells in the placenta/decidua (Fig S7). This result has been added in the manuscript (page 9, lines 220-223 and page 11, lines 285-287).

5. The ANOVA test with Tukey's multiple comparison tests are commonly used for a normal (Gaussian) distribution. Did the author's test for normality of the data? Or did they transform the data? The authors should test for normality of data distribution (for example using histograms, stem-and-leaf plots, or normal probability plots or hypothesis tests, such as the Kolmogorov-Smirnov and Shapiro-Wilk tests) to know whether to use a parametric or non-parametric test. Data analyzed by nonparametric statistics are commonly presented as the median along with an appropriate range (minimum and maximum values, upper and lower quartiles or quintiles, etc.) and not use mean \pm SEM. The following articles are useful for statistics:

Martin Krzywinski & Naomi Altman. Points of significance: Nonparametric tests. Nature Methods 11, 467-468 (2014). PMID: 24820360.

Cara H. Olsen. Statistics in Infection and Immunity Revisited. Infect Immun. 2014 Mar; 82(3): 916-

920. PMCID: PMC3957975

Following the reviewer's recommendation, we tested normality of the data using JASP software (version 0.8.6, 2018) and found that most of the data was not a normal (Gaussian) distribution. Consequently, for morphological measurements and viral burden data, a Kruskal-Wallis test was used to compare more than two groups. Viral burden data and ICS data were analyzed using a two-tailed Mann-Whitney test to compare two groups. Percentages of infection in placenta with decidua and fetal tissues were assessed via two-sided Fisher's exact test. For morphological measurements and viral burden, data was expressed as a median. ICS data was shown as an interquartile range. These changes are reflected in the new version of the manuscript and figures.

Reviewer #2 (Remarks to the Author):

The major claim of the paper is that cross-reactive CD8 cells (against Dengue) may protect against Zika virus. However, the main premise is not fully tested. Maternal CD8 cells (placental infiltrates) are known to be associated with adverse perinatal outcomes as in cases of intrauterine inflammation (e.g., Lei et al., 2017). Therefore, the title and the premise of the manuscript may deal only with viral transmission but not necessarily the adverse perinatal outcomes associated with CD8 cells. As is the manuscript is well-written but lacks translational potential.

Lei et al. previously investigated the role of CD8 T cells in perinatal brain injury using a mouse model of intrauterine inflammation in which dams were administered LPS via intrauterine injection at E17, and they demonstrated that maternal CD8 T cell depletion reduced LPS-induced perinatal brain injury¹⁶. In contrast, our study focuses on evaluating the contribution of viral antigen-specific CD8 T cells in protecting against or pathogenesis of ZIKV infection early during pregnancy. Our new gain-of-function studies (peptide immunization and adoptive transfer) demonstrating that cross-reactive virus-specific CD8 T cells can control ZIKV infection at the maternal-fetal interface argue that the intrauterine LPS injection model and our model of systemic ZIKV infection during pregnancy represent two different phenomena (i.e., generalized decidual inflammation where antigen-non-specific CD8 T cells are pathogenic and pathogen-specific infection and injury where antigen-specific CD8 T cells contribute to reduced viral burden and protection). We have mentioned the Lei et al study finding in the Discussion (page 11, lines 290 and 291).

References

- 1 Miner, J. J. et al. Zika Virus Infection during Pregnancy in Mice Causes Placental Damage and Fetal Demise. *Cell* 165, 1081-1091, doi:10.1016/j.cell.2016.05.008 (2016).
- 2 Yockey, L. J. et al. Vaginal Exposure to Zika Virus during Pregnancy Leads to Fetal Brain Infection. *Cell* 166, 1247-1256 e1244, doi:10.1016/j.cell.2016.08.004 (2016).
- 3 Yockey, L. J. et al. Type I interferons instigate fetal demise after Zika virus infection. *Sci Immunol* 3, doi:10.1126/sciimmunol.aao1680 (2018).

- 4 *Plummer, E. & Shresta, S. Animal models in dengue. Methods Mol Biol 1138, 377-390, doi:10.1007/978-1-4939-0348-1_23 (2014).*
- 5 *Plummer, E. M. & Shresta, S. Mouse models for dengue vaccines and antivirals. J Immunol Methods 410, 34-38, doi:10.1016/j.jim.2014.01.001 (2014).*
- 6 *Yauch, L. E. & Shresta, S. Mouse models of dengue virus infection and disease. Antiviral Res 80, 87-93, doi:S0166-3542(08)00328-8 [pii] 10.1016/j.antiviral.2008.06.010 (2008).*
- 7 *Zellweger, R. M. & Shresta, S. Mouse models to study dengue virus immunology and pathogenesis. Front Immunol 5, 151, doi:10.3389/fimmu.2014.00151 (2014).*
- 8 *Onlamoon, N. et al. Dengue virus-induced hemorrhage in a nonhuman primate model. Blood 115, 1823-1834, doi:10.1182/blood-2009-09-242990 (2010).*
- 9 *Silveira, E. L. V. et al. Immune Cell Dynamics in Rhesus Macaques Infected with a Brazilian Strain of Zika Virus. J Immunol 199, 1003-1011, doi:10.4049/jimmunol.1700256 (2017).*
- 10 *Szaba, F. M. et al. Zika virus infection in immunocompetent pregnant mice causes fetal damage and placental pathology in the absence of fetal infection. PLoS Pathog 14, e1006994, doi:10.1371/journal.ppat.1006994 (2018).*
- 11 *Nishiura, H. Duration of short-lived cross-protective immunity against a clinical attack of dengue: A preliminary estimate. WHO Regional Office for South-East Asia. Dengue Bulletin. 2008; 32: 55-66. (2008.).*
- 12 *Reich, N. G. et al. Interactions between serotypes of dengue highlight epidemiological impact of cross-immunity. J R Soc Interface 10, 20130414, doi:10.1098/rsif.2013.0414 (2013).*
- 13 *SABIN., A. B. Research on dengue during World War II. . Am J Trop Med Hyg. 1:30 –50. (1952.).*
- 14 *Anderson, K. B. et al. A shorter time interval between first and second dengue infections is associated with protection from clinical illness in a school-based cohort in Thailand. J Infect Dis 209, 360-368, doi:10.1093/infdis/jit436 (2014).*
- 15 *Snow, G. E., Haaland, B., Ooi, E. E. & Gubler, D. J. Review article: Research on dengue during World War II revisited. Am J Trop Med Hyg 91, 1203-1217, doi:10.4269/ajtmh.14-0132 (2014).*
- 16 *Lei, J. et al. Maternal CD8(+) T-cell depletion alleviates intrauterine inflammation-induced perinatal brain injury. Am J Reprod Immunol 79, e12798, doi:10.1111/aji.12798 (2018).*

REVIEWERS' COMMENTS:

Reviewer #1 (Remarks to the Author):

The authors have addressed the major criticisms of the reviewers.

Reviewer #2 (Remarks to the Author):

My concerns were addressed.

Response to Reviewers: NCOMMS-18-01114A

“Cross Reactive Dengue Virus Specific CD8+ T Cells Protect Against Zika Virus During Pregnancy”

REVIEWERS' COMMENTS:

Reviewer #1 (Remarks to the Author):

The authors have addressed the major criticisms of the reviewers.

We thank the reviewer and greatly appreciate this positive comment.

Reviewer #2 (Remarks to the Author):

My concerns were addressed.

We appreciate the reviewer's comment.